# Detecting Vulnerabilities in Critical Infrastructures by Classifying Exposed Industrial Control Systems Using Deep Learning



Pablo Blanco-Medina [1,2,*], Eduardo Fidalgo [1,2], Enrique Alegre [1,2], Roberto A. Vasco-Carofilis [1,2], Francisco Jañez-Martino [1,2] and Victor Fidalgo Villar [2]

1   Department of Electrical, Systems and Automation, Universidad de León, 24071 León, Spain; eduardo.fidalgo@unileon.es (E.F.); enrique.alegre@unileon.es (E.A.); rvasc@unileon.es (R.A.V.-C.); fjanm@unileon.es (F.J.-M.)
2   INCIBE (Spanish National Cybersecurity Institute), 24005 León, Spain; victor.fidalgo@incibe.es
*   Correspondence: pblanm@unileon.es

**Featured Application: We present a deep-learning-based pipeline to solve a novel problem in Cybersecurity and Industry 4.0. Our proposal, which automatically classifies screenshots of industrial control systems, might support the task of an industrial monitoring tool for detecting vulnerable or exposed industrial control systems on the internet, which might be related to critical infrastructures.**

**Abstract:** Industrial control systems depend heavily on security and monitoring protocols. Several tools are available for this purpose, which scout vulnerabilities and take screenshots of various control panels for later analysis. However, they do not adequately classify images into specific control groups, which is crucial for security-based tasks performed by manual operators. To solve this problem, we propose a pipeline based on deep learning to classify snapshots of industrial control panels into three categories: internet technologies, operation technologies, and others. More specifically, we compare the use of transfer learning and fine-tuning in convolutional neural networks (CNNs) pre-trained on ImageNet to select the best CNN architecture for classifying the screenshots of industrial control systems. We propose the critical infrastructure dataset (CRINF-300), which is the first publicly available information technology (IT)/operational technology (OT) snapshot dataset, with 337 manually labeled images. We used the CRINF-300 to train and evaluate eighteen different pipelines, registering their performance under CPU and GPU environments. We found out that the Inception-ResNet-V2 and VGG16 architectures obtained the best results on transfer learning and fine-tuning, with F1-scores of 0.9832 and 0.9373, respectively. In systems where time is critical and the GPU is available, we recommend using the MobileNet-V1 architecture, with an average time of 0.03 s to process an image and with an F1-score of 0.9758.

**Keywords:** deep learning; image classification; transfer learning; industrial control system; fine-tuning

## 1. Introduction

Interconnection between electronic devices connected to the internet has become necessary to ensure the control, communication, and monitoring of multiple systems. Those systems, which are exposed online, should be deployed under various security measures to avoid potential attacks [1].

In critical infrastructures, such as healthcare, transportation, or manufacturing, a system shutdown or restart would lead to severe economic and social consequences, as well as significant time costs [2]. Furthermore, the threat of a potential security breach can range from an information leak to system overtaking, which entails high risks in environments

such as industrial control systems (ICSs) [1]. Due to this, these systems must rely on constant surveillance [1] to guarantee robustness and stability.

Supervisory control and data acquisition (SCADA) systems are used to control physical equipment and ICS infrastructures. SCADA systems are commonly referred to as operational technology (OT) systems, which directly control and monitor specific devices. Other industrial systems used to control software, including management, storage, and delivery of data, are known as information technology (IT) systems [3].

To monitor these exposed assets, law enforcement agencies (LEAs) use open-source intelligence (OSINT) tools [4]. In particular, specialized tools, such as Shodan [5], known as metasearchers, monitor the open ports of a network, and the services of the devices that are exposed to the internet. For services that include a graphical user interface (GUI), specific metasearchers usually take screenshots to log relevant information graphically.

The classification of these assets is useful for determining the types of compromised devices. Therefore, the screenshots taken help to discover vulnerabilities, classify the devices based on the images taken, and analyze the obtained information afterwards. However, these metasearchers may not correctly classify images as belonging to IT or OT, thus requiring a manual classification. Due to the large number of devices connected to the internet and the multiple monitoring options existing in the metasearchers, this manual process can be an arduous task for a human operator. Moreover, the dynamic environment and continual updates of these systems may increase the difficulty of classifying these images.

In our work, we propose a pipeline based on existing deep learning models to solve a novel problem related to cybersecurity. More specifically, we use convolutional neural networks (CNNs) pre-trained on an ImageNet dataset to automatically classify screenshots of ICSs that could be linked to critical infrastructures. The ICS snapshots are taken during the monitoring of open ports and devices exposed to the internet through OSINT sources. We designed and evaluated several pipelines based on transfer learning and fine-tuning, which first take and resize a screenshot of an ICS as input. On the one hand, in the transfer learning approach, we use a pre-trained CNN architecture for extracting features. Then, they are fed to a classifier trained with images labeled in three categories, i.e., OT, IT, and others. On the other hand, we fine-tuned the same set of CNN architectures to automatically classify the three categories mentioned above. We consider architectures used in similar studies [6], but we also included others available in the field of computer vision.

We trained and evaluated our pipelines in the critical infrastructure classification dataset (CRINF-300), a dataset of 337 samples crawled from Shodan and manually labeled with IT and OT categories. Even if it is a small dataset for image classification, to the best of our knowledge, it is the first to contain snapshots of real and exposed IT and OT systems. The requirement for a large amount of training data is alleviated due to transfer learning and data augmentation techniques capable of constructing a robust and powerful classifier [7]. Thanks to our collaboration with the Spanish National Cybersecurity Institute (INCIBE) (https://www.incibe.es/en), we applied the pipeline that we present in this paper to recognize industrial control systems based on screenshots in order to support the task of detecting vulnerable systems exposed on the internet in real-time.

The rest of the paper is organized as follows. Section 2 presents a summary of the state-of-the-art image classification approaches and architectures. In Section 3, we introduce the methodology followed. Section 4 discusses our experimental settings and the obtained results. Lastly, in Section 5, we present our conclusions and future lines of work.

## 2. State of the Art

Image classification is the task of assigning a label to an image. Traditionally, handcrafted features [8] were extracted from the images and used for training classifiers. CNNs have been established among the best learning algorithm for image-based tasks [9] by achieving the best results on the ILSVRC (ImageNet Large-Scale Visual Recognition Challenge) [10].



Despite this, there are cases where the number of images for training a model is scarce, or the classification tasks are challenging. In these cases, manually crafted feature extraction can outperform the results obtained by CNNs [11,12].

Their parameter optimization and their capability of changing their structures to fit different problems [9,13] have allowed the improvement of CNNs' performance over the years. Technological advances like the use of graphics processing units (GPUs) have also made their progress possible [14].

As seen in [15], CNNs can be divided into seven different categories according to their architecture: spatial exploitation, feature-map exploitation, depth, width, multi-path, channel boosting, and attention. Multiple networks can appear in various categories.

However, these networks need to be trained on a large amount of data, and data gathering and annotation can be a complex, tedious, and time-consuming process. Furthermore, these datasets may soon become outdated, needing the addition of new data [16].

Transfer learning is a technique that allows one to take a model trained for a specific application and apply it to a similar or related task [17]. This approach is used to retain the features obtained from bigger datasets and use them for training a new model on a smaller, similar dataset. Several works have studied the use of transfer learning applied to CNNs for the task of image classification in different fields, but often omit the most recent architectures [6,18], such as NasNetLarge [19]. A brief summary of the most notable image classification CNNs can be seen in Table 1.

**Table 1.** Image classification architectures' accuracy scores.

| Architecture | Top-5 Accuracy (%) | Top-1 Accuracy (%) | Dataset |
|---|---|---|---|
| LeNet [20] | 99.80 | - | MNIST [20] |
| DenseNet [21] | 93.88 | 77.85 | CIFAR-10 [22] |
| AlexNet [23] | 84.60 | 63.30 | ImageNet |
| ZFNet [24] | 84.00 | 62.50 | ImageNet |
| GoogleNet [25] | 89.90 | 69.80 | ImageNet |
| VGG16 [26] | 91.90 | 74.40 | ImageNet |
| ResNet [27] | 94.29 | 78.57 | ImageNet |
| ResNeXt-101 [28] | 95.60 | 80.90 | ImageNet |
| Inception-V3 [29] | 94.40 | 78.80 | ImageNet |
| SENet [30] | 96.20 | 82.70 | ImageNet |
| MobileNet-V1 [31] | 90.92 | 71.56 | ImageNet |
| MobileNet-V2 [32] | - | 74.70 | ImageNet |
| MobileNet-V3 [33] | - | 75.20 | ImageNet |
| EfficientNet [34] | 97.00 | 84.30 | ImageNet |
| Xception [35] | 94.50 | 79.00 | ImageNet |
| Inception-ResNet-V2 [36] | 95.10 | 80.10 | ImageNet |
| NasNetLarge [19] | 96.20 | 82.70 | ImageNet |

Hussain et al. [17] studied the application of transfer learning on the InceptionV3 [29] architecture pre-trained on the ImageNet dataset [37] and re-trained it on the CIFAR-10 [22] dataset, obtaining 70.1% accuracy and surpassing CNNs trained from scratch on this dataset.

Sharma et al. [38] applied transfer learning on AlexNet [23], GoogLeNet [25], and ResNet50 [27]. They replaced the last three layers with a fully connected layer, a softmax layer, and a classification output layer for each network. Afterwards, they trained the networks on the CIFAR-10 dataset, obtaining classification accuracy per image category. The resulting average performance for each network was 71.67% for GoogLeNet, 78.10% for ResNet50, and 36.12% for AlexNet.

Extensive architecture research and engineering are required to improve neural network classification [39]. To fix this problem, Bello et al. [40] proposed an approach called

neural architecture search that helps in optimizing the architecture configuration and improving the classification performance and training time on the CIFAR-10 dataset.

However, training architectures on large datasets, such as ImageNet, causes techniques like this to have a high computational cost. Therefore, Zoph et al. [19] proposed the use of a smaller dataset, CIFAR-10, as a proxy, and then transferred the learned architecture to the ImageNet dataset. The resulting architecture, called NASNet, was compared to multiple CNNs, such as MobileNet-224 [31], Inception-ResnetV2 [36], and Xception [35], on image classification on the ImageNet dataset, comparing both the number of parameters and the accuracy. The proposed solution achieved state-of-the-art results on the ImageNet dataset.

## 3. Methodology

### 3.1. Critical Infrastructure Dataset

For our proposal, we used ICS images provided by the Spanish National Cybersecurity Institute (INCIBE), which were retrieved using multiple metasearchers, such as Shodan (https://www.shodan.io). We manually labeled the 337 snapshots in two categories: 74 IT and 263 OT images, and we named the resulting dataset the Critical Infrastructure (CRINF-300). Although our small dataset may be insufficient for training a classifier from scratch, we used data augmentation techniques to increase the number of training images by five times when we performed fine-tuning and transfer learning, which allowed us to properly train a robust classifier. Figure 1 presents four screenshots of IT and OT systems to appreciate their differences.

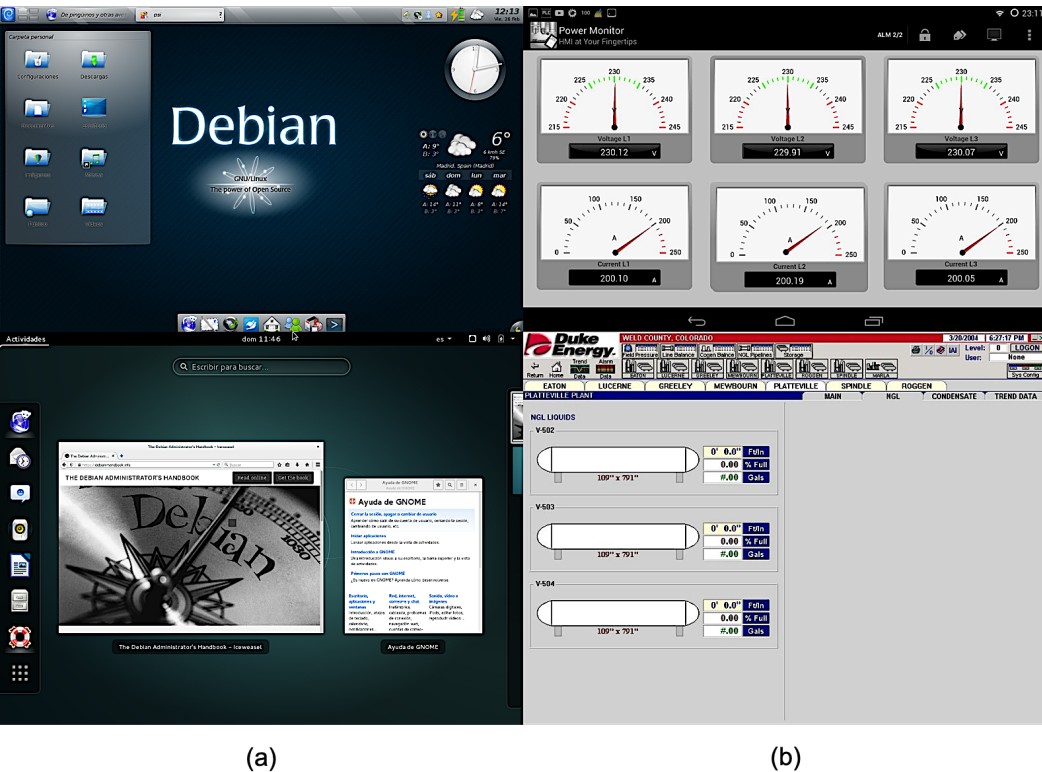

(a)                                                                (b)

**Figure 1.** Information technology (IT) (column **a**) and operational technology (OT) (column **b**) sample images. IT systems focus on software management and data control, while OT systems help directly monitor device values.

### 3.2. Proposed Pipeline

We modified and evaluated several pipelines based on transfer learning and fine-tuning, which use resized screenshots of ICSs as input. On the one hand, in transfer learning, we used a pre-trained CNN architecture for extracting features, which were then fed to a classifier trained in three image categories: OT, IT, and others. On the other hand, we performed fine-tuning on a newly defined head of the model of the CNN

architecture to automatically classify an image into the three previous categories. Using nine different CNNs, we trained three classifiers applying both transfer learning and fine-tuning techniques. After obtaining the classifier, we used it to label images in three categories: IT, OT, and others, according to the classifier's confidence score. If it was below a certain threshold, 0.9 in our experiments, the images were classified as Others. This third category helps identify additional clusters retrieved from the metasearchers, such as Internet of Things (IoT) images, which could be useful for adding future labels. For our experiments with the selected architectures, we only classified images as IT or OT, omitting the third category for performance purposes. Figure 2 presents an overview of the proposed systems for classifying screenshots.

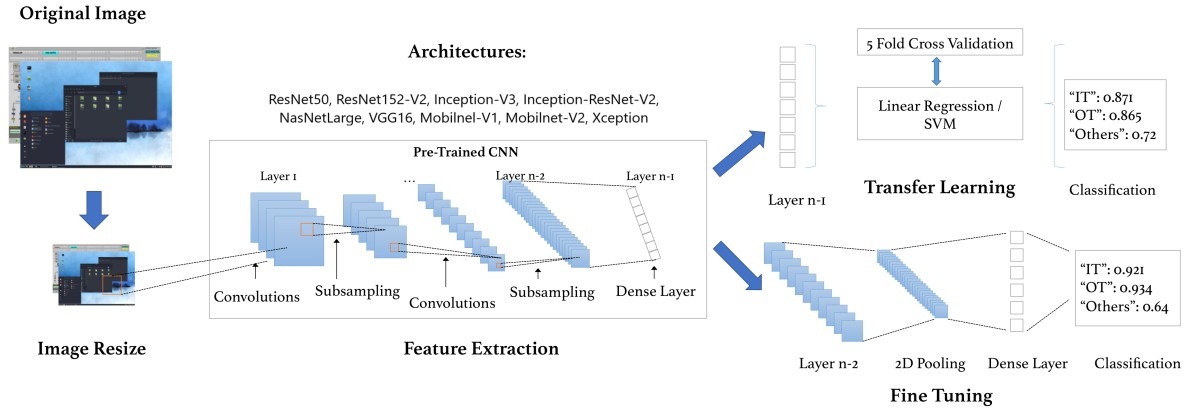

**Figure 2.** Transfer learning and fine-tuning pipeline. Transfer learning and fine-tuning were used to train classifiers and output the final labels.

### 3.2.1. Transfer Learning and Fine-Tuning

Due to the limited number of images available, our initial approach was focused on the implementation of transfer learning, instead of re-training the architectures from scratch. For the experiments, we selected a total of nine architectures to apply to our image classification problem: Inception-V3 [29], MobileNet-V1 [31], MobileNet-V2 [32], ResNet50, ResNet152v2 [27], VGG16 [26], NasNetLarge [19], Inception-ResNet-V2 [36], and Xception [35].

We froze the final layer for each network using the pre-trained ImageNet weights and obtained the features before classification. Then, we used those features to train two separate classifiers based on a logistic regression model and a support vector machine (SVM) with a linear kernel, which would output the final labels.

In addition to transfer learning, we also propose fine-tuning in the previously selected architectures for our classification problem based on their F1-score.

### 3.2.2. Architectures

We selected the given architectures due their common use in similar problems that use transfer learning, and computer vision techniques [18,39]. Additionally, specific architectures, such as MobilenetV1, were chosen due to the real-time-based nature of the given task, which focuses on mobile, lightweight deployment.

VGG [26] is an architecture focused on spatial exploitation, which attempts to consider various filter sizes to analyze both low- and high-level details [15]. It is known for its vast number of parameters, which complicates deployment and increases computational costs, but achieves good results in image classification and object localization tasks.

ResNet [27] introduced the concept of residual learning in CNNs by creating connections between layers that speed up network convergence. It comprises 152 layers, eight times more than the original VGG, while being less complicated. Due to these connection types and its focus on depth, it is classified as both depth- and multi-path-based. The network variations using 50 and 101 layers, respectively, have shown remarkable results in

tasks of image classification [15]. ResNet152v2's [27] main difference when compared to the original is that it uses batch normalization before each of the weight layers.

Inception-V3 [29] is a depth- and width-based CNN focused on reducing the computational cost of these types of networks. However, it has a sophisticated design and lacks homogeneity. Inception-ResNet-V2 [36] was introduced as a higher-cost variant of the Inception network, with a higher object recognition performance.

Xception [35] attempts to regulate computational complexity, making learning more efficient and improving the architecture's performance, but has the disadvantage of a high computational cost compared to other architectures.

NasNetLarge [19] designs a search space to reduce architecture complexity and make it independent of image size or network depth, finding the best possible network on a smaller dataset before scaling it to bigger datasets. The resulting architecture can be further modified to fit a larger scale, achieving state-of-the-art results.

Lastly, MobileNet-V1 [31] is an architecture oriented towards mobile device applications, and is designed to have a light architecture for ease of deployment. MobileNet-V2 [32] improves the original proposal by introducing linear bottlenecks between the layers and shortcut connections between them, reducing operation complexity and the parameters needed.

## 4. Experimental Results and Discussion

### 4.1. Experimental Settings

We evaluated our proposal on an Intel Xeon E5 v3 computer with 128 GB of RAM using an NVIDIA Titan Xp GPU for both training and testing. All of the CNN architectures were implemented using Python3 under the Keras library [41] and using Tensorflow as the backend.

To measure the performance of both our approaches, we used the F1-score and the accuracy metrics, as can be seen in Equations (3) and (4). We chose these metrics to represent the robustness of our classifiers. The F1-score is the harmonic mean of the precision and recall measures, which are detailed in Equations (1) and (2), respectively. In these equations, TP and FP represent true and false positives, respectively, while TN and FN indicate true and false negatives.

$$P = \frac{TP}{TP + FP} \tag{1}$$

$$R = \frac{TP}{TP + FN} \tag{2}$$

$$F1 = 2 \cdot \frac{P \cdot R}{P + R} \tag{3}$$

$$Accuracy = \frac{TP + TN}{TP + TN + FP + FN} \tag{4}$$

In image classification problems, a true positive is when an image is assigned its correct label. For a particular class, a false positive is considered for each image that has been labeled as belonging to a given class, but belongs to a different one. At the same time, false negatives are considered as the images from the class that have been assigned other labels.

We selected the 0.9 value for the confidence threshold of the pipeline empirically. Following the requirements of the monitoring service where the proposal would be integrated, we initially evaluated high threshold values of $-0.8$ and $0.9-$ over a small set of images. We visually inspected the results of the classification under both thresholds and concluded that 0.9 obtains a higher generalization capability with a lower number of failures, meeting the requirements of the monitoring tool.

We also measured the time needed for feature extraction and classification in both the CPU and GPU, retrieving both the mean and standard deviation in each task.

### 4.1.1. Transfer Learning Settings

To fit the architecture's input size, each image was resized to the required value. For VGG16, ResNet50, and both MobileNet architectures, images were fixed to a size of $224 \times 224$. On the Xception, ResNet152v2, and Inception-based architectures, they were scaled to $299 \times 299$, while NasNetLarge used a size of $331 \times 331$. We fed the resized images to these pre-trained networks, extracted the features, and trained our classifiers using them. Both the logistic regression classifier and the SVM classifier were implemented using the Scikit-learn Python library [42].

Since there is a difference between the available images per category, we implemented five-fold cross-validation, generating five models per architecture. This technique helps reduce model bias when compared to other approaches, such as the training–testing split. The dataset was divided into five folds, four of which were used to fit the model, whereas the last one is used for validation. This process was repeated until every fold was used to test the proposed model.

### 4.1.2. Fine-Tuning Settings

Using the previously selected architectures, we performed fine-tuning by dropping the last layer of each model, adding a new set of layers that consisted of an average 2D pooling of $7 \times 7$, a Flatten layer, a Dense layer with ReLu activation and 256 outputs, a 0.5 dropout layer, and a final Dense layer with softmax activation and two outputs.

For fine-tuning, we used 70% of our images for training, 20% for testing, and 10% for validation. For a fair comparison, we added the same layers with the same parameters on all the architectures. We trained these added layers with a batch size of 26 images and a learning rate of $1 \times 10^{-4}$ over 20 epochs, we evaluated them with the same set of images. To increase the number of images for training our architectures, we used data augmentation in order to learn more general and robust features, improving the performance of our models.

We used the ImageDataGenerator function from Keras to create these new images. We set the rotation range parameter to 25, zoom range to 0.1, width shift range to 0.1, height and shift range to 0.1, shear range to 0.2, horizontal flip to True, and fill mode to "nearest". These new slightly altered images were used for training and consisted of five times the number of images of the original training batch.

### 4.2. Discussion of Results

The results of our transfer learning experiments can be seen in Tables 2 and 3. In both of the proposed approaches, Inception-ResNet-V2 obtained the highest F1-score and accuracy. Using the logistic regression classifier, this architecture obtained a 98.32 F1-score and 97.33 accuracy, with a variance of 0.70 and 1.12, respectively. On the SVM classifier, Inception-ResNet-V2 obtained slightly lower results, with a 98.13 F1-score and 97.03 accuracy. VGG16 and ResNet50 obtained the lowest F1-score, with 87.67 in both approaches. ResNet152v2 obtained a 10% F1-score over ResNet50, but with a computational time three times higher, highlighting the enhancements of the batch normalization.

NasNetLarge was the slowest-performing architecture in all our given approaches in both the CPU and GPU, which can be attributed to the large size of the network. When combining transfer learning with an SVM classifier, NasNetLarge took an average of 0.54 s in the CPU and 0.27 s in the GPU to process a single image. In the CPU, it was followed by ResNet152v2 with 0.33 s and 0.17 s in the GPU. NasNetLarge took an average of 0.49 s in the CPU and 0.26 s in GPU to process a single image, the fastest time recorded for this architecture in all of our approaches. In the CPU, it was followed by ResNet152v2 with 0.49 s and by Inception-ResNet-V2 in the GPU with 0.12 s.

**Table 2.** F1-score and accuracy results of our transfer learning strategy using a logistic regression classifier on critical infrastructure classification dataset (CRINF-300) with five-fold cross-validation. The time indicates the average computational cost per individual image. Bold highlights the best results.

| Architecture | F1-Score (%) | Accuracy (%) | CPU (s) | GPU (s) |
|---|---|---|---|---|
| ResNet50 | 87.67 **(+/−0.35)** | 78.04 **(+/−0.55)** | 0.10 (+/−0.10) | 0.05 (+/−0.18) |
| VGG16 | 87.67 **(+/−0.35)** | 78.04 **(+/−0.55)** | 0.16 **(+/−0.03)** | **0.03 (+/−0.07)** |
| Xception | 89.46 (+/−0.67) | 81.60 (+/−1.24) | 0.14 (+/−0.07) | 0.05 (+/−0.15) |
| Inception-V3 | 97.02 (+/−1.50) | 95.25 (+/−2.40) | 0.10 (+/−0.16) | 0.07 (+/−0.26) |
| Mobilenet-V1 | 97.58 (+/−0.70) | 96.15 (+/−1.17) | **0.06** (+/−0.05) | 0.04 (+/−0.12) |
| Mobilenet-V2 | 97.55 (+/−0.49) | 96.14 (+/−0.73) | 0.08 (+/−0.09) | 0.05 (+/−0.17) |
| NasNetLarge | 96.44 (+/−1.29) | 94.36 (+/−1.99) | 0.49 (+/−0.65) | 0.16 (+/−0.96) |
| Inception-ResNet-V2 | **98.32** (+/−0.70) | **97.33** (+/−1.12) | 0.21 (+/−0.41) | 0.12 (+/−0.63) |
| ResNet152v2 | 97.41 (+/−0.90) | 95.85 (+/−1.46) | 0.26 (+/−0.32) | 0.10 (+/−0.46) |

**Table 3.** F1-score and accuracy results of our transfer learning strategy using a support vector machine (SVM) classifier on CRINF-300 with five-fold cross-validation. The time indicates the average computational cost per individual image. Bold highlights the best results.

| Architecture | F1-Score (%) | Accuracy (%) | CPU (s) | GPU (s) |
|---|---|---|---|---|
| ResNet50 | 87.67 **(+/−0.35)** | 78.04 **(+/−0.55)** | 0.10 (+/−0.11) | 0.06 (+/−0.18) |
| VGG16 | 87.67 **(+/−0.35)** | 78.04 **(+/−0.55)** | 0.14 **(+/−0.03)** | **0.03 (+/−0.07)** |
| Xception | 90.08 (+/−1.51) | 82.77 (+/−2.83) | 0.14 (+/−0.07) | 0.05 (+/−0.14) |
| Inception-V3 | 97.20 (+/−1.31) | 95.54 (+/−2.12) | 0.14 (+/−0.16) | 0.12 (+/−0.26) |
| Mobilenet-V1 | 97.56 (+/−0.75) | 96.14 (+/−1.19) | **0.08** (+/−0.05) | 0.06 (+/−0.11) |
| Mobilenet-V2 | 97.92 (+/−0.74) | 96.73 (+/−1.12) | 0.10 (+/−0.09) | 0.07 (+/−0.16) |
| NasNetLarge | 96.78 (+/−1.04) | 94.96 (+/−1.52) | 0.54 (+/−0.66) | 0.27 (+/−0.81) |
| Inception-ResNet-V2 | **98.13** (+/−0.84) | **97.03** (+/−1.33) | 0.27 (+/−0.40) | 0.15 (+/−0.55) |
| ResNet152v2 | 97.59 (+/−0.94) | 96.14 (+/−1.52) | 0.33 (+/−0.32) | 0.17 (+/−0.42) |

The MobileNet architecture obtained the second-best performance. On the logistic approach, V1 obtained scores of 97.58 and 96.15 against the 97.55 and 96.14 obtained by V2. When using the SVM classifier, MobileNet-V2 yielded slightly better results, with a 97.92 F1-score and 96.73 accuracy compared to the 97.56 and 96.14 results reported by V1. Despite these slight differences, MobileNet-V1 always achieved faster CPU and GPU times than V2, which included less variance in the time required to process the images.

When comparing the logistic regression approach against the SVM classifier, we saw small improvements in both the F1-score and accuracy in Xception, InceptionV3, NasNet-Large, ResNet152V2, and MobileNet-V2. However, both MobileNet-V1 and Inception-ResNet-V2 obtained slightly lower F1 results, decreasing to 97.56 and 98.13, respectively. Lastly, ResNet50 and VGG16 remained with the same scores performance-wise.

As for performance variance, ResNet50 and VGG16 obtained the least variance in both approaches for F1-score and accuracy, with 0.35 and 0.55, respectively. In our logistic classifier, InceptionV3 obtained the largest variance of F1-score and accuracy with 1.50 and 2.40, respectively, followed by NasNetLarge with 1.29 and 1.99. On our SVM approach, the Xception architecture surpassed the variance of these two models, with 1.51 F1-score and 2.83 accuracy variances.

Regarding the computational cost, VGG16 and MobileNet-V1 were the best-performing architectures in both types of transfer learning. MobileNet-V1 was faster on the CPU, with 0.06 s against the 0.16 obtained by VGG16, but was surpassed on the GPU by 0.03 s.

On average, the SVM classifier took slightly more time than the logistic regression approach, with the highest average increase seen in the NasNetLarge architecture from 0.49 to 0.54 on the CPU and 0.16 to 0.27 on the GPU. While the differences in most architectures

between the two proposed transfer learning approaches are minimal, they can be critical in real-time systems.

Our fine-tuned results are presented in Table 4. VGG16 obtained the best results with an F1-score of 93.73 and an accuracy of 95.59, which improved significantly from the transfer learning results of 87.67 F1-score and 78.04 accuracy. ResNet50 was the second-best-performing architecture, with a 92.16 F1-score and 94.12 accuracy.

**Table 4.** Results using the fine-tuning strategy over the nine selected architectures, using the F1-score and accuracy. The time indicates the average computational cost per individual image. Bold highlights the best results.

| Architecture | F1-Score (%) | Accuracy (%) | CPU (s) | GPU (s) |
|---|---|---|---|---|
| ResNet50 | 92.16 | 94.12 | 0.27 (+/−0.25) | 0.20 (+/−0.25) |
| VGG16 | **93.73** | **95.59** | **0.21 (+/−0.07)** | **0.14 (+/−0.10)** |
| Xception | 80.98 | 88.24 | 0.23 (+/−0.08) | 0.15 (+/−0.12) |
| Inception-V3 | 82.01 | 88.24 | 0.28 (+/−0.18) | 0.18 (+/−0.17) |
| MobileNet-V1 | 86.51 | 91.18 | **0.21 (+/−0.14)** | 0.18 (+/−0.19) |
| MobileNet-V2 | 48.78 | 76.47 | 0.25 (+/−0.19) | 0.18 (+/−0.15) |
| NasNetLarge | 81.20 | 83.82 | 0.68 (+/−0.44) | 0.32 (+/−0.43) |
| Inception-ResNet-V2 | 76.22 | 85.29 | 0.36 (+/−0.17) | 0.22 (+/−0.27) |
| ResNet152v2 | 71.22 | 83.82 | 0.34 (+/−0.15) | 0.19 (+/−0.22) |

While ResNet50 and VGG16 obtained significant improvements in both F1-score and accuracy, they did not surpass the best results yielded by the transfer learning approach. Furthermore, they increased the computational time required by up to three times when compared to the transfer-based models. However, VGG16 remained as the architecture with the least time variance and fastest processing time, followed by MobileNet-V1 on the CPU and Xception on the GPU. These enhanced scores suggest that fine-tuning of the layers and parameters is more beneficial to these architectures than simply freezing the final layers, despite our low number of images.

Although our fine-tuning additions perform well on top of ResNet50 and VGG16, they do not achieve good performance on the rest of the architectures, obtaining lower results than those of the transfer-learning-based models. In particular, MobileNet-V2 obtained an F1-score of 48.78 from the original 97.55. This drop in performance can also be attributed to our new head of the model and the low number of training images.

Given the real-time processing requirement of monitoring ICSs and the particular focus on light deployment, we recommend the pipeline with the Mobilenet-V1 architecture in order to have the best speed–performance trade-off. Although VGG16 is the fastest architecture on the GPU for our fine-tuning approach, and with the least variance across all proposals, its performance is much lower than that of Mobilenet-V1 in transfer learning. Figures 3 and 4 graphically illustrate our results, highlighting the best-performing architectures across all three approaches.

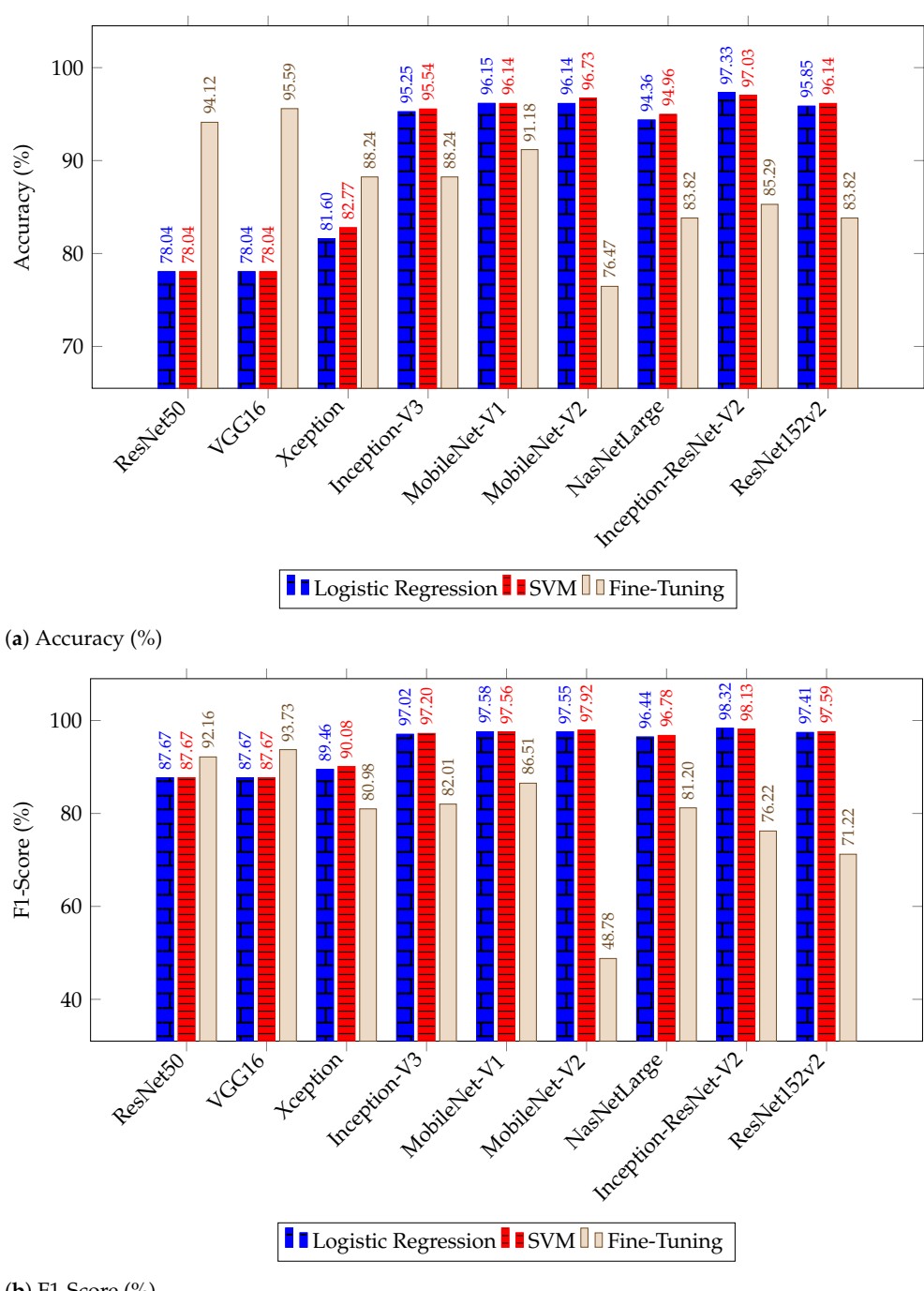

(**a**) Accuracy (%)

(**b**) F1-Score (%)

**Figure 3.** Accuracy (**a**) and F1-score (**b**) comparison of the results obtained in our proposed pipeline using transfer learning and fine-tuning on CRINF-300.

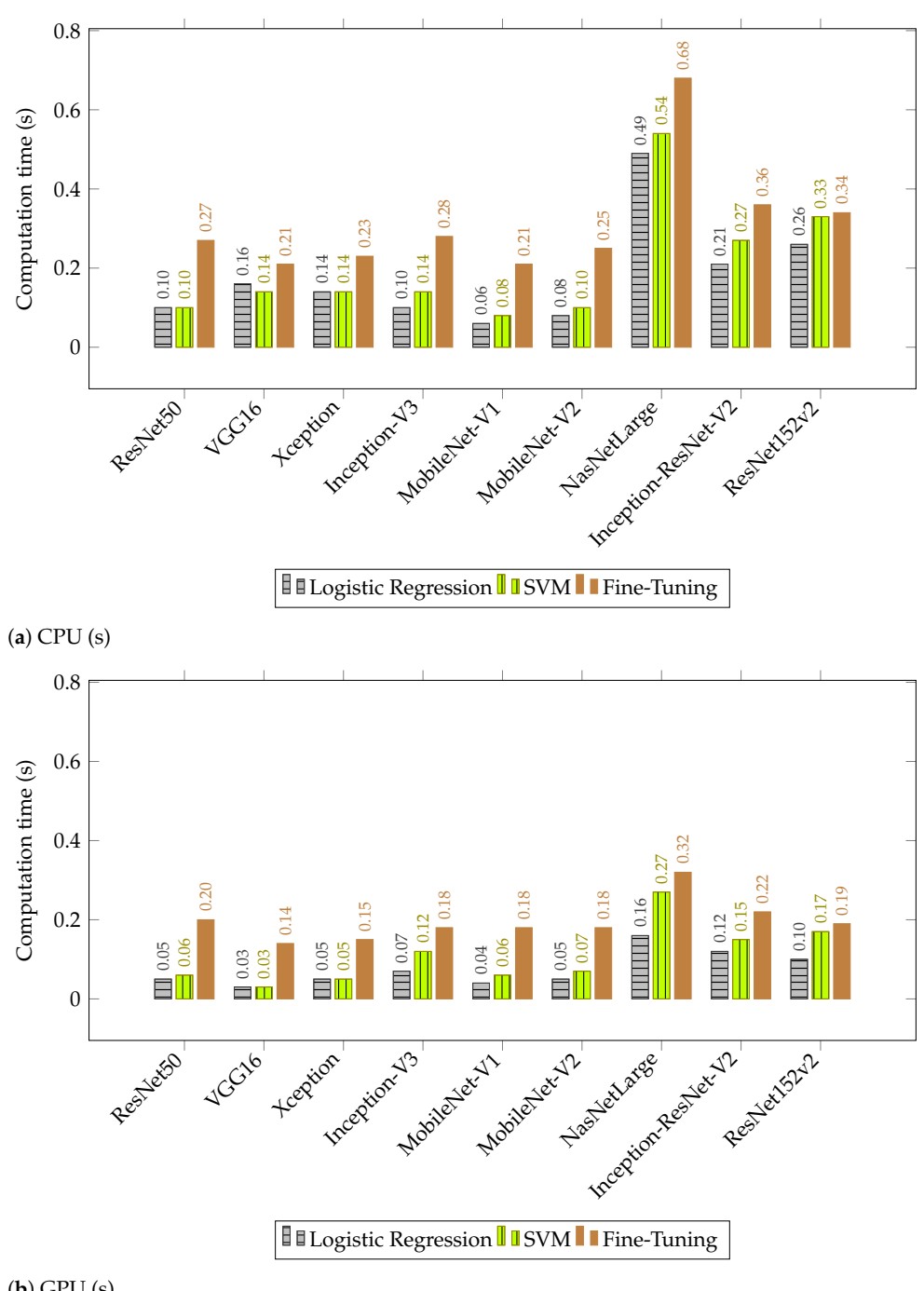

(**a**) CPU (s)

(**b**) GPU (s)

**Figure 4.** Time comparison of the results obtained in our proposed pipeline using the CPU (**a**) and the GPU (**b**).

## 5. Conclusions

This paper has presented a pipeline for classifying SCADA images as belonging to IT or OT systems using transfer-learning- and fine-tuning-based approaches on a custom SCADA dataset named CRINF-300. Moreover, we have applied it to a real-case scenario where we classify images to enhance surveillance tasks on critical infrastructure systems to detect and prevent potential security breaches.

We have analyzed nine different CNN architectures using their pre-trained weights on the ImageNet dataset to train three different approaches for image classifiers: transfer

learning with a logistic regression classifier or with an SVM classifier and fine-tuning. We also register the average time in the classification of each image on both the CPU and GPU.

For transfer learning, we validated our approach using the F1-score, and accuracy, and five-fold cross-validation during training. We trained and tested the architectures on a 337 image dataset provided by INCIBE, containing 74 IT images and 263 OT images.

We proposed a new model head after freezing the last layers of the selected architectures and adding a custom set of layers for fine-tuning. We enhanced our images with data augmentation and trained our model using training–testing validation.

Our results show that the best CNN architectures for our problem are Inception-ResNet-V2, as it is the best architecture performance-wise with a 98.32 and 98.13 F1-score on our transfer-learning-based approach, and MobileNet-V1, as it has the best performance–speed trade-off, with an F1-score of 97.58 and a speed of 0.06 s on the CPU. Out of the two classifiers proposed for transfer learning, although close in performance, the logistic regression model seems to be better suited for our problem, with higher average scores, reduced performance variance, and decreased computational time.

We conclude that transfer learning is the better approach for our problem based on the performance of the Inception-ResNet-V2 and MobileNet-V1 networks. While VGG16 obtained a noticeable improvement when fine-tuning the architecture, it did not surpass MobileNet's original score, even when increasing our total number of images by up to five times using data augmentation.

Our future work will be focused on further improving the proposed solution, increasing our training images, and adding new layers on top of the given architectures based on our VGG16 results. Another possibility is to extend the current study further to include different architectures, such as VGG19 [26]. The inclusion of more detailed classes is also a future line of investigation for this image classification task.

Finally, after classification is performed, images can be further analyzed by retrieving information found within them [8], such as brand or company names, using text spotting [43,44].

**Author Contributions:** Conceptualization, P.B.-M., E.F., and E.A.; methodology, P.B.-M., E.F., and E.A; software, P.B.-M.; validation, P.B.-M., E.F., and E.A.; formal analysis, P.B.-M., E.F., E.A., R.A.V.-C., F.J.-M., and V.F.V.; investigation, P.B.-M.; resources, E.F. and E.A.; data curation, P.B.-M.; writing—original draft preparation, P.B.-M.; writing—review and editing, P.B.-M., E.F., E.A., R.A.V.-C., F.J.-M., and V.F.V.; visualization, P.B.-M.; supervision, E.F. and E.A.; project administration, E.F. and E.A.; funding acquisition, E.A. All authors have read and agreed to the published version of the manuscript.

**Funding:** This research was supported by the grant "Ayudas para la realización de estudios de doctorado en el marco del programa propio de investigación de la Universidad de León Convocatoria 2018" and by the framework agreement between Universidad de León and INCIBE (Spanish National Cybersecurity Institute) under Addendum 01.

**Institutional Review Board Statement:** Not applicable.

**Informed Consent Statement:** Not applicable.

**Data Availability Statement:** Data sharing is not applicable in this article.

**Acknowledgments:** This work was supported by the framework agreement between the Universidad de León and INCIBE (Spanish National Cybersecurity Institute) under Addendum 01. We acknowledge the NVIDIA Corporation for the donation of the TITAN Xp and Tesla K40 GPUs used for this research.

**Conflicts of Interest:** The authors declare no conflict of interest. The funders had no role in the design of the study.

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
