# Peer review of "Detecting Vulnerabilities in Critical Infrastructures by Classifying Exposed Industrial Control Systems Using Deep Learning"

_applsci, doi:10.3390/app11010367_

Round 1

Reviewer 1 Report

In order to examine the exposure of industrial control systems, the author conducted a study to automatically classify screenshots of open ports and devices exposed to the outside through OSINT sources. The author proposed a pipeline based on Deep Learning to classify snapshots of images into 3 categories, IT, OT, and Others. Image classification models were trained using CNN, one of the deep learning models. The 18 pipelines were evaluated with 337 images crawled by Shodan.
The proposed method is necessary because the risk of cyber-attacks is increasing these days. However, it has the following problems.

1. The problem to be solved in this paper is not clear.
The paper analyzed the experimental results by applying several existing techniques. However, there is a mention that a new method is proposed in the introduction. So the problem to solve this paper is not clear. The author needs to clarify whether this paper focused on proposing a novel method or applying a proper existing model.
2. A detailed description of the proposed technique is needed.
The author used Transfer Learning with several classifiers. Each of them is already used in various fields. It is necessary to emphasize what the authors proposed.
3. 337 images are used for validating the performances of pipelines. Since the amount of data is too little, the author should be mentioned the reason why it is sufficient for training and testing.

Reviewer 2 Report

See the attachment.

Round 2

Reviewer 1 Report

It seems to have been modified enough to be published.